# Tumor-Associated Microbiome: Where Do We Stand?

**DOI:** 10.3390/ijms22031446

**Published:** 2021-02-01

**Authors:** Marc Oliva, Nuria Mulet-Margalef, Maria Ochoa-De-Olza, Stefania Napoli, Joan Mas, Berta Laquente, Laia Alemany, Eric J. Duell, Paolo Nuciforo, Victor Moreno

**Affiliations:** 1Medical Oncology Department, Catalan Institute of Oncology L’Hospitalet de Llobregat, 08908 Catalonia, Spain; nmulet@iconcologia.net (N.M.-M.); blaquente@iconcologia.net (B.L.); 2ONCOBELL Program, Bellvitge Biomedical Research Institute (IDIBELL), L’Hospitalet de Llobregat, 08908 Catalonia, Spain; jmas@idibell.cat (J.M.); eduell@idibell.onmicrosoft.com (E.J.D.); v.moreno@iconcologia.net (V.M.); 3Service of Immuno-Oncology, Department of Oncology, Lausanne University Hospital, 1011 Lausanne, Switzerland; Maria.Ochoa-De-Olza@chuv.ch; 4Ludwig Institute for Cancer Research, University of Lausanne, 1066 Lausanne, Switzerland; 5Molecular Oncology Group, Vall d’Hebron Institute of Oncology, 08035 Barcelona, Spain; snapoli@vhio.net (S.N.); pnuciforo@vhio.net (P.N.); 6Oncology Data Analytics Program, Catalan Institute of Oncology (ICO), L’Hospitalet de Llobregat, 08908 Catalonia, Spain; 7Consortium for Biomedical Research in Epidemiology and Public Health (CIBERESP), 28029 Madrid, Spain; lalemany@iconcologia.net; 8Cancer Epidemiology Research Program, Catalan Institute of Oncology, L’Hospitalet de Llobregat, 08908 Catalonia, Spain; 9EPIBELL Program, Bellvitge Biomedical Research Institute (IDIBELL), L’Hospitalet de Llobregat, 08908 Catalonia, Spain

**Keywords:** tumor microbiome, gut microbiome, dysbiosis, cancer, carcinogenesis, metagenomics

## Abstract

The study of the human microbiome in oncology is a growing and rapidly evolving field. In the past few years, there has been an exponential increase in the number of studies investigating associations of microbiome and cancer, from oncogenesis and cancer progression to resistance or sensitivity to specific anticancer therapies. The gut microbiome is now known to play a significant role in antitumor immune responses and in predicting the efficacy of immune-checkpoint inhibitors in cancer patients. Beyond the gut, the tumor-associated microbiome—microbe communities located either in the tumor or within its body compartment—seems to interact with the local microenvironment and the tumor immune contexture, ultimately impacting cancer progression and treatment outcome. However, pre-clinical research focusing on causality and mechanistic pathways as well as proof-of-concept studies are still needed to fully understand the potential clinical utility of microbiome in cancer patients. Moreover, there is a need for the standardization of methodology and the implementation of quality control across microbiome studies to allow for a better interpretation and greater comparability of the results reported between them. This review summarizes the accumulating evidence in the field and discusses the current and upcoming challenges of microbiome studies.

## 1. Introduction: Microbiology Meets Oncology

This is the “decade of microbiome”, reported Forbes’ last publication of 2019. While the existing link between microbiome and health in the human host has been known for years, it was not until recently that the influence of the microbiome reached several medical disciplines, including oncology, going from unknown to mainstream.

The human microbiome is defined as the collective genomes and by-products of all the microorganisms inhabiting the human body, including bacteria, viruses, fungi, protozoa, and archaea [1,2]. These microbial communities are distributed in multiple compartments throughout the body (e.g., skin, oro-gastrointestinal, urogenital tracts) and vary in type and number depending on each compartment, although they all seem to share core functions such as glycolysis, ATP synthesis, and activation of translational machinery [1,2]. The microbiome has a symbiotic and dynamic relationship with the human host, with some microorganisms being key players in several physiological functions that mainly involve regulation of metabolic processes and immune system responses [2]. Although the precise underlying mechanisms are not completely understood, changes in the microbiome composition of a specific body compartment caused by either host intrinsic or external factors (e.g., genetics, infections, diet, or antibiotics) can alter the local homeostasis and induce chronic inflammation, damaging tissues and dysregulating local and systemic immune responses, ultimately leading to disease [3]. These pathology-associated alterations are also known as dysbiosis, and they have been linked to several disorders, including cancer [4]. 

The association between the microbiome and cancer is not new. Up to 20% of cancers are actually related to infections [5], and several pathogenic bacteria and viruses contribute to the etiopathogenesis of specific tumor types [6,7,8]. However, beyond these well-established agent-tumor causality associations, overall quantitative and/or qualitative shifts in the microbiome composition of a specific compartment may also trigger cancer initiation, development, and progression. The International Cancer Microbiome Consortium postulates that the microbiome is one apex of a carcinogenesis-leading tripartite, jointly with (epi)genetics and environment [9]. This idea is supported by the Ecological Koch’s postulate that sustains that a dysbiosis resulting from (epi)genomics, environment, and microbiome leads to a single disease [10].

In addition to its role in tumorigenesis and cancer progression, the microbiome has also emerged as a new potential biomarker in cancer diagnosis, risk stratification, and prognosis. Microbial signatures detected in cell-free DNA from human fluids have been linked to specific tumor types and could be used for diagnostic purposes [11]. Other studies have shown a correlation between tumor-associated bacteria and survival or response to anticancer therapies [12,13]. Recently, accumulating evidence has implicated the gut microbiome in the modulation of host anticancer immune responses and the efficacy of immune-checkpoint inhibitors across many tumor types [14,15,16,17]. These findings have led to preclinical and clinical investigations on how to manipulate the microbiome to use it as a therapeutic tool to boost the efficacy of anticancer therapies through different strategies, from dietary interventions and probiotic/antibiotic therapies to fecal microbial transplantation [3,18] (NCT04264975, NCT01895530, NCT03817125). 

## 2. Tumor-Associated Microbiome

Bacteria, viruses, and other micro-organisms located in different body compartments have been correlated with increased susceptibility of developing different cancers [19,20,21,22]. Cancer patients seem to harbor a specific microbiome composition in the tumor niche and also within the tumor’s body compartment, which differs from healthy controls [23,24,25,26]. These specific changes in the microbial communities intratumor or nearby observed in cancer patients are what we define as tumor-associated microbiome. Whether this tumor-associated microbiome is involved in carcinogenesis or if it is merely a by-stander effect due to the tumor microenvironment is yet to be fully elucidated. It is hypothesized that a dysbiosis in a specific compartment or tissue could start an oncogenic process through (1) the induction of chronic inflammation, (2) the inhibition of cellular apoptosis, (3) the production and release of carcinogenic substances, or (4) the modulation of local anti-tumor immunity and tumor microenvironment [27]. For instance, changes in the relative abundance (RA) of a given group of bacteria has been shown to directly cause DNA damage leading to genetic dysregulation and initiation of tumorigenesis [28,29]. A recent elegant study conducted by Nejman et al. revealed that intra-tumor microbiome composition is diverse and cancer type-specific [23]. Interestingly, bacteria found in tumor tissue were biologically active and mainly located in the cytoplasm of both tumor and immune cells, suggesting an implication in both oncogenesis and antitumor immunity.

In this section, we revise the evidence available on the tumor-associated microbiome by cancer type and its potential clinical use as a diagnostic, prognostic, or predictive biomarker (Table 1).

### 2.1. Cancers of the Upper Aerodigestive Tract: Head and Neck and Esophageal Tumors

Head and neck, esophageal, and gastric cancers arise from the epithelium and mucosa of the oro-gastrointestinal tracts and upper airway. These compartments are constantly exposed to external aggressions such as smoking, alcohol consumption, or infections, which can alter their microbiome composition [62,63]. Some viral infections such as Epstein–Barr virus (EBV) and human papillomavirus (HPV) are well-established etiopathological agents of nasopharyngeal (NPC) and oropharyngeal carcinomas (OPC), respectively [60,64], while the bacterial species *Helicobacter pylori* (*H. pylori*) is causally associated with the incidence of gastric adenocarcinomas and mucosa-associated lymphoid tissue lymphoma [6,65]

However, beyond these specific pathogen–tumor type causality relationships, which will be further discussed, oro-gastrointestinal dysbiosis has been correlated with increased risk of head and neck (HNC), esophageal (EC), and gastric cancers (GC) [66,67,68,69]. Several retrospective case–control studies have found differential microbiome composition in the saliva, mucosal, and tumor tissues of patients with these tumor types when compared to healthy individuals, suggesting an implication in tumor initiation and development [25,70,71,72]. Other commensal bacteria have been shown to be protective of cancer development and could be used for cancer prevention purposes—both *Kingella* and *Corynebacterium* species, which are functionally implicated in the biodegradation and/or metabolization of carcinogens from tobacco and/or alcohol (e.g., Acetaldehyde), have been linked with decreased risk of head and neck squamous cell carcinomas (HNSCC) among smokers/alcohol consumers in a nested case–control study within a prospective cohort [20,30].

Oropharyngeal and esophagogastric compartments share similar commensal microorganisms as well as microbiome pathogenic alterations [73]. For instance, increased RA of oral *Porphyromonas gingivalis*, a bacterium associated with periodontal disease, has been suggested to facilitate the development of oral carcinomas through the activation of immune evasion mechanisms and oncogenic pathways, but it also has been correlated with cancer cell differentiation and metastasis in patients with esophageal squamous cell carcinoma (SCC) [31,74]. *Fusobacterium nucleatum (F. nucleatum)* is also found in both head and neck and esophageal SCC and is associated with advanced tumor stages and a more aggressive tumor behavior in both patient populations [32,33,75]. In contrast, different microbiome composition has been described within the same compartment in association with a specific tumor histology, indicating that intercompartmental dysbiosis might lead to different tumor types or vice versa; whether these findings are cause or consequence is yet unclear. *Campylobacter* species are found increased in the esophageal mucosa of patients with gastroesophageal reflux disease (GERD) and Barrett’s esophagus, and seem to be implicated in the development of esophageal adenocarcinomas but not SCC through the activation of immune pathways linked to toll-like receptors [34,35,36]. In the case of EC and GC, infections by other microorganisms such as fungi have also been implicated in carcinogenesis through mucosal injury and dysregulation of the local immune system and oncogenic pathways [76,77].

Fewer studies are available on the potential impact of tumor-associated microbiome on outcome and response to therapy in patients with cancer of the upper aerodigestive tract. In HNC, there is no evidence of oral/tumor-associated microbiome as a biomarker of response to standard therapies such as radiotherapy, chemotherapy, or immunotherapy, although studies are underway (NCT03410615). To date, the only study that evaluated the oral microbiome in a subgroup of patients with recurrent/metastatic HNSCC treated with antiPD-1 agent nivolumab within the CheckMate-141 clinical trial failed to show any correlation with treatment response [78]. However, the small number of patients and the low percentage of responses might have influenced these results. In terms of toxicity, two studies have shown a correlation between oral dysbiosis and increased radiation-induced mucositis in patients with HNC [79,80]. In patients with esophageal SCC, increased RA of intratumoral *F. nucleatum* has been associated with poor response to neoadjuvant chemoradiation and higher risk of recurrence [13].

### 2.2. Hepatocarcinoma, Pancreas, and Biliary Tract Cancers

Cancers from the hepato-biliary system are under the influence of the microbiomes belonging to each of the organs involved but also of the gut microbiome via blood flow through the portal vein [37]. The relationships between gut microbiome, biliary acids, and liver diseases, including hepatic steatosis, non-alcoholic liver disease, non-alcoholic steatohepatitis, cirrhosis, biliary tract cancers, and hepatocellular carcinoma (HCC), have been reviewed extensively [81,82]. A recent pre-clinical study by Zhang et al. evaluated the gut microbiome in 127 mouse models for primary sclerosis cholangitis, colitis, and cholangiocarcinoma [83]. They were able to show that Gram-negative commensal bacteria from the gut control the accumulation of hepatic myeloid-derived immunosuppressive cells (MDSCs) through a TLR4/CXCL1/CXCR2129-dependent mechanism and thus contribute to an immune-suppressive microenvironment in the liver [83].

Among viruses, hepatitis B (VHB) and C (VHC) infections are well-established risk factors not only for liver cancer but also for pancreatic ductal adenocarcinoma (PDAC). Alcohol-induced tumors (including HCC and PDAC) were observed to have distinct microbiome composition from virally induced tumors, suggesting that liver microbiome may differ in response to different etiological factors [84]. Beyond viruses, certain pathogenic bacteria such as *H. pylori* and oral periopathogens such as *P gingivalis*, *Fusobacterium* sp., *Aggregatibacter* sp., *Prevotella* sp., or *Capnocytophaga* sp. seem to play a role in the development of PDAC via induction of chronic inflammation, antiapoptotic changes, cell survival, and cell invasion [37]. In this regard, a study by Pushalkar et al. detected specific gut and tumor microbiome in murine models of PDAC, suggesting a potential bacterial translocation from the intestinal tract into the peritumoral milieu [85]. Interestingly, PDAC-associated microbiome as well as gut microbiome were involved in immune-suppression in pancreatic tissue, a characteristic often observed in PDAC. Together, these data suggest that gut and/or tumor microbiome represent a potential therapeutic target to modulate disease progression in PDAC.

The PDAC-associated microbiome appears to also have a prognostic role, although its correlation with the incidence of this disease has not been evaluated sufficiently. Riquelme et al. evaluated the intratumor microbiome composition of PDAC patients according to short-term survival (STS) and long-term survival (LTS), identifying a specific intra-tumoral microbiome signature (*Pseudoxanthomonas*–*Streptomyces*–*Saccharopolyspora*- *Bacillus clausii*) that was predictive of long-term survivorship in both discovery and validation cohorts [26]. Chakladar et al. profiled the intra-tumor pancreatic microbiome through large-scale sequencing data from The Cancer Genome Atlas (TCGA) (187 pancreatic cancer samples). The authors found that the increased prevalence and poorer prognosis of PDAC in males and smokers were linked to the presence of potentially cancer-promoting or immune-inhibiting microbes (most of them belonged to Proteobacteria phylum) [86]. Another study showed that intra-tumor Gammaproteobacteria in PDAC modulates tumor sensitivity to gemcitabine, one of the few active and standard of care chemotherapy drugs used in PDAC [38].

In HCC, a small study evaluated the changes in gut microbiome after antiPD-1 therapy in eight patients with Barcelona Clinic Liver Cancer (BCLC) Stage C disease. Differences in microbiome diversity and composition were observed between responders and non-responders, thus suggesting that gut microbiome dynamics might be predictive of response to these agents in patients with HCC [87].

### 2.3. Colorectal Cancer

The microbiome in colorectal cancer (CRC) is one of the most studied across malignancies, but its role in the development of this tumor is still a matter of debate. The “Driver-Passenger CRC model” defines as drivers those bacteria with pro-carcinogenic features that are found in pre-malignant lesions or in early CRC, while the term “passengers” refers to bacteria that act as tumor promoters or suppressors in later stages of disease [88]. Among drivers, enterotoxigenic *Bacteroides fragilis* drives tumor growth through different mechanisms encompassed in the so-called alpha-bugs hypothesis, such as DNA damage, induction of cell proliferation, and induction of T helper 17 inflammation [44]. *Escherichia coli,* which is a producer of toxin colibactin *(pk+)*, has also raised interest as a driver, since it may cause toxin-induced DNA damage, promoting a specific CRC mutational profile based on insertions and deletions [45]. *F. nucleatum* is the paradigmatic passenger bacterium because it is rarely detected in adenoma, but it may have a relevant role at latter stages of carcinogenesis [89]. Preclinical studies have shown that this species is capable of activating oncogenic pathways such as MAPK and Wnt [19,90], and to impair antitumor immune response through the activation of NF-κB signature and the interaction with immune-checkpoints [91,92,93]. Of note, its presence has been also found in synchronous or metachronous liver metastases from CRC primary tumors harboring this bacterium [94], suggesting that *F. nucleatum* could disseminate to other organs/locations via systemic circulation, such as cancer cells. Moreover, *F. nucleatum* is more abundant in right-sided tumors [95] and those with mismatch repair deficiency, indicating a potential relationship with the mutator phenotype pathway of CRC carcinogenesis [39,40].

Beyond the oncogenic role of the abovementioned species, the CRC-associated microbiome has emerged as a potential screening tool as well as a prognostic and predictive biomarker. The detection of a specific bacterial signature (including *Peptostreptococcus stomatis, Parvimonas* spp., and *Porphyromonas* sp., among others) in stools may be used for screening purposes on the basis of the results of two meta-analysis of seven and eight datasets whose patients belonged to different geographic areas, including Europe, Asia, and North America [96,97]. Higher levels of *F. nucleatum* in CRC tissue correlated with worse disease-specific survival in the largest series with more than 10 years of follow-up [41]. The persistence of this same species in tumor tissue after neoadjuvant chemoradiotherapy for locally advanced rectal cancer was associated with higher relapse rates, while other studies have shown a correlation between higher levels of the bacteria and resistance to oxaliplatin and 5-fluorouracil in the adjuvant setting [12,42,43].

Beyond bacteria, the composition of other microbiota such as viruses, fungi, and archaea seem to be different in CRC but their direct impact in CRC carcinogenesis or their utility in tumor management are still unknown [46,47,48].

### 2.4. Genitourinary Cancers

Genitourinary cancers are a miscellany of tumors whose data regarding their tumor-associated microbiome is scarcer than in other malignancies. Like stool in CRC, urine must also be considered in the study of microbiome associated with kidney cancer and urothelial carcinoma. In spite of the postulated sterility of urine, very preliminary data obtained through sequencing methods suggest the presence of bacteria in the urine of healthy individuals [98].

A few studies have shown differential urine microbiome composition in patients with urothelial carcinoma when compared to healthy controls, mainly characterized by an enrichment of *Fusobacterium* and Firmicutes and a decrease of *Streptococcus* RA [49,50]. However, the potential causality relationship between the bladder tissue/urine microbiome and urothelial carcinoma—most frequent histology in bladder cancer—is yet to be elucidated. The only exception is schistosomiasis as a well-established cause of the squamous carcinoma of the bladder, but as a result of previous infection by this pathogen [99]. In renal cell carcinoma, different taxonomic profiles consistent in higher RA of *Chloroplast* and *Streptophyta* have been described in the tumor niche when compared to surrounding normal tissue [51]. A prognostic role of urine/tissue microbiome has not been described, neither in urothelial carcinoma nor in kidney cancer.

In prostate adenocarcinoma, intratumor bacteria such as *Listeria monocytogenes* have been found to be inversely correlated with adverse prognostic features (Tumor-Node-Metastasis classification, Gleason score, prostate serum antigen, levels, or androgen receptor expression) and it is hypothesized that they counteract tumor growth via local recruitment of immune cells [53]. Moreover, some other intratumor bacteria seem to correlate with specific genomic alterations associated with tumor progression and local immune suppression. From a therapeutic perspective, *Akkermansia muciniphila* seems to be relevant for the activity of abiraterone acetate in patients with castrate-resistant prostate cancer. This bacterium triggers the bacterial biosynthesis of vitamin K2, which inhibits androgen-dependent tumor growth [52]. 

The contribution of genital tract microbiome in the pathogenesis of female genital-tract malignancies is also raising interest. Ovarian cancer tissue samples associate a specific microbiome profile of fungi, viruses, parasites, and bacteria [100]. In the same way, endometrial cancer shows higher representation of *Porphyromonas* sp. and *Atopobium vaginae* compared with healthy tissue [54].

### 2.5. Other Cancers

**HPV- and EBV-related cancers:** both HPV and EBV are known to initiate the oncogenic process through viral DNA integration into the human genome and through acquisition of cell survival capabilities, causing different tumors depending on the body compartment or organ infected [64,101]. HPV is a well-established cause of oropharyngeal and anogenital tract squamous cell carcinomas, while EBV is directly related with nasopharyngeal and gastric cancers as well as some types of lymphoma [60,101,102,103]. Beyond the etiopathogenic role of these agents, they also seem to impact the composition of the tumor-associated microbiome. The group of Guerrero-Preston reported different prevalence and RA of specific taxa between HPV-related and unrelated oropharyngeal carcinomas [32]. Interestingly, this study also observed that the saliva of patients with HPV-related oropharyngeal carcinomas was found to be enriched in commensal species (*Lactobacillus* species) from the vaginal flora. In this regard, changes in the composition of the vaginal microbiome have been associated with the risk and clearance of HPV infection as well as with development of pre-malignant cervical lesions [59]. However, the mechanisms involved in these correlations have not been elucidated. Beyond vaginal fluid, stool samples from patients with localized cervical cancer showed different microbiome composition when compared to healthy controls. This brings about the possibility of using stools as a diagnostic tool for early-stage cervical cancer and, in fact, preliminary data have shown good performance in differentiating healthy patients from cancer patients according to gut microbiome profile using stool samples [104].

Data on the role of microbiome in EBV-related cancers are scarce. EBV-associated gastric carcinomas account for nearly 10% of gastric cancers [101]. A recent study involving a very small number of patients was able to detect differences in gut microbiome composition between EBV-related and unrelated carcinomas [61]. The gut bacterial functional pathways using the Kyoto Encyclopedia of Genes and Genomes data and tumor expression of immune-lipid metabolism functional proteins by immunohistochemistry (IHC) differed in terms of EBV presence as well. A score based on these factors was found to be predictive of outcome in this cancer [61]. Whether tumor-associated microbiome has a prognostic or predictive role in terms of response to therapies has not yet been evaluated.

**Breast ductal carcinomas** have different microbiome composition when compared to adjacent normal tissue and overlying skin within the same patient, and also when compared to breast tissue from healthy individuals [58,105]. Interestingly, intratumor taxonomic composition of breast cancer patients appear to differ also according to the tumor subtype (triple-negative vs. triple-positive ductal carcinomas).

In **lung cancer**, many studies have consistently reported different bacterial communities in the lung tissue of patients with lung cancer when compared to healthy individuals [22,106,107]. A meta-analysis of epidemiologic studies analyzed previous lung infections as risk factors for lung cancer. The results showed that a previous infection by *Chlamydia pneumoniae* or *Mycobacterium tuberculosis* was associated with an increased risk of lung cancer [55]. Although the potential mechanisms between the microbiome and lung carcinogenesis are not well-known, it seems that the metabolites produced by certain bacteria might be potentially oncogenic [56]. In that sense, pre-clinical in vitro and in vivo research from Tsay et al. showed that exposure to *Veillonella*, *Prevotella*, and *Streptococcus* bacteria are capable of inducing epithelial cell transformation through the activation of the PI3K and ERK pathways [56].

## 3. Microbiome and Antitumor Immunity

### 3.1. Interplay between the Microbiome, the Immune System, and Response to Anticancer Therapies

The crosstalk between gut microbiome and the immune system is key to maintain the intestinal homeostasis as it enables tolerance to commensal microorganisms while inducing inflammatory responses against invading pathogens. These gut microbiome interactions are in fact crucial for shaping and modulating innate and adaptive immune responses locally and also systemically, as they are responsible for the development and maturation of myeloid and lymphoid cells [108,109,110,111]. Gut microbial communities can balance immune responses towards an anti- or pro-inflammatory effect, depending on the type of immune cell they affect [112]—specific bacteria and their by-products (metabolites) have anti-inflammatory effects by inducing T regulatory cell differentiation [113,114,115], while other are pro-inflammatory as they activate/stimulate dendritic cells (DC), T helper cells, or CD8+ cells [116,117,118,119,120]. Multiple mechanisms orchestrate this microbiome–immune system crosstalk [121]. For example, microbial-associated molecular patterns (MAMPs) from gut bacteria are detected by toll-like receptors (TLR) and can directly modify the function and maturation of innate immune cells [122]. Additionally, metabolites produced by certain bacteria such as trimethylamine N-oxide (TMAO) [123] and butyrate [124] can modulate innate immune cell differentiation and polarization [121]. Hence, the gut microbiome not only contributes to the immune system development, but also balances pro- and anti-inflammatory immune cell responses, ultimately having an effect on a variety of diseases such as cancer, auto-immune diseases, and obesity [125].

The interplay between the gut microbiome and the immune system can also affect antitumor immune-mediated responses (Figure 1) [109]. Accumulating data indicates that tumor responses to chemotherapies such as gemcitabine [38] and cyclophosphamide [126] depend on the gut microbiome. Several studies have shown a correlation between the gut microbiome composition and diversity and the efficacy of immunotherapy in patients with different tumor types, including melanoma, renal clear cell carcinoma, and lung cancer [4,15,17,127,128,129,130]. Recent data from melanoma patients revealed that the administration of stools from responders to immune checkpoint inhibitors (ICI) to non-responders can revert the primary resistance to these agents and lead to increased tumor infiltration by CD8 T cells [131], as previously suggested in pre-clinical studies [127]. Although further research is warranted, these data indicate an existing link between gut microbiome composition and tumor immune responses in cancer patients. Although the underlying mechanisms explaining this correlation are still not fully understood, a few hypotheses have been suggested [132,133]. One of the hypotheses is that some antigens are shared between bacteria and tumors and thus lead to cross-reactive T cells against the tumor cells. In this regard, recent data involving non-small cell lung cancer (NSCLC) and renal cell carcinoma (RCC) patients showed that the expression of an enterococcal cross-reactive antigen by tumors correlated with response to anti-PD-1 therapy [134]. Other proposed mechanisms include T cell priming and activation mediated by dendritic cells upon presentation of microbe- and pathogen-associated molecular patterns (MAMPs and PAMPs, respectively) present in the gut or from systemic circulation or increased pro-inflammatory cytokines and microbial metabolites [16,133,135,136,137,138]. Zhang et al. recently demonstrated that gut bacteria induce the expression of immunosuppressive chemokines in hepatocytes that cause the accumulation of MDSCs, ultimately promoting the development and growth of cholangiocarcinomas [83].

Beyond the gut, the tumor-associated microbiome might also play a role in antitumor immune responses, although less data are available in this regard [13,38,85]. Unraveling the exact mechanisms through which gut- and tumor-associated microbiome can mediate antitumor immune-responses will be crucial in order to tailor microbiome manipulation to boost antitumor responses in cancer patients.

### 3.2. Modulation of Gut Microbiome to Boost Antitumor Responses

Preclinical and clinical studies strongly support the key role of the gut microbiome in the modulation of systemic and antitumor immune responses in cancer patients [139]. However, many host intrinsic and extrinsic factors such as genetic susceptibility, dietary habits, or concurrent medication contribute to the microbiome composition and diversity and might ultimately affect immune-mediated antitumor responses [140,141]. For example, antibiotics are a known cause of gut dysbiosis [4], and their use seems to detrimentally impact on the overall survival and progression-free survival of cancer patients [142,143,144], and also impair responses to ICI [145,146].

The therapeutic manipulation of the gut microbiome to increase the efficacy of anticancer therapies, particularly of immunotherapy, is under evaluation, and several strategies have been proposed including dietary modifications; the use of probiotics, prebiotics, or selected antibiotics; and fecal microbiota transplantation (FMT) [18,147,148]. A recent review on this specific topic discusses the advantages and disadvantages of each of these approaches and highlights some on-going trials [149].

Dietary changes such as including or excluding specific nutrients classes (e.g., lipids) or diet supplementation with oral probiotics or prebiotics are capable of altering the gut microbiome composition [150]. Probiotics are “live organisms that might confer a health benefit to the host” while prebiotics are dietary fibers that are non-digestible by the host but digestible by gut microbes, and as such, they can favor the colonization and expansion of particular bacteria and their specific metabolites [151]. The combination of prebiotics and probiotics is known as synbiotics [152]. Pre-clinical studies suggest that diet and pre- and probiotics can enhance immune response and have antitumor properties via several mechanisms including modulation of apoptosis and cell differentiation, production of pro-inflammatory cytokines (IL-2, IL-12, and IFN-y), antioxidants (superoxide dismutase, catalase, glutathione peroxidase), and anti-angiogenic factors and reduction of cancer-specific proteins, polyamine contents, and pro carcinogenic enzymes [153,154]. However, whether they actually may enhance antitumor responses and boost the efficacy of therapies in cancer patients is still unknown.

Other strategies such as FMT, that is, a fecal suspension into the digestive tract, or stool substitutes such as oral bacterial consortia (mixture of pure live cultures of bacteria, often isolated from a stool sample of a healthy donor) are promising [155]. FMT has been proven successful for recurrent and refractory *Clostridium difficile* infection and has rapidly expanded to multiple fields of extra-gastrointestinal diseases [156,157]. Recently, Baruch et al. performed a phase I clinical trial to assess the safety, feasibility, and immune cell impact of FMT plus anti-P-D1 in PD-1 in refractory metastatic melanoma patients. Interestingly, this combination appeared safe and induced radiological tumor responses and tumor immune infiltration by CD8+T cells [131].

## 4. Microbiome in Oncology: Are We Ready for Prime Time?

A recent report from the International Agency for Cancer Research points out a high degree of heterogeneity across microbiome studies in terms of method of description, techniques used, taxonomic deepness, and lack of information about confounding factors [22]. There is an urge for standardization of methodology and result-reporting as well as for bias control in microbiome-related studies. Figure 2 summarizes the current challenges of microbiome studies in cancer.

### 4.1. Benchmarks in Standardization of Collection and Preservation Methods

Microbiome analyses can be performed in multiple types of biological samples (e.g., tumor tissue, body fluids, or stools) and using different collection and preservation methods, and as such, results obtained might vary. For instance, while gut bacterial communities seem to have a homogeneous distribution along the colon mucosa, the overall diversity of the microbiome differs when we use stool samples versus intestinal mucosal tissue [158]. The choice of sample type and collection and storage methods when studying tumor-associated microbiome is highly relevant. While it might be obvious that microbiome composition will differ between separated body compartments (e.g., oral vs urinary tract), it is unclear whether tumor tissue or a sample from the cancer-associated compartment (e.g., oral cancer tissue vs. saliva or CRC tissue vs. stool) would be equally representative. Stool samples currently used for gut microbiome analysis are limited if the goal is to study CRC-associated microbial communities [159]. In contrast, in HNC studies, microbiome composition, and diversity appeared similar when using saliva, tumor tissue, or tumor swab [33]. The same has been shown in patients with urothelial cancer when using urine and tumor tissue [160]. However, more studies to further evaluate this are needed.

Sample handling and preservation methods are relevant in order to avoid bacterial continuous growth and contamination. Several studies have analyzed the variability and the stability of microbiome diversity and composition when using different times and/or preservation temperatures [161]. In general, immediate sample freezing at -20ºC is considered the best option, but this may not be always feasible. In regard to stool samples, preservation using 95% ethanol, fecal occult blood test (FOBT), fecal immunochemical tests (FIT) tubes, Flinders Technology Associates (FTA) cards, or RNAlater provide good stability at room temperature up to 7 days, showing good correlation with fresh frozen samples [162]. No preservation media or 70% ethanol are not recommended. The International Human Microbiome Standards (IHMS) consortium (http://www.microbiome-standards.org) has published guidelines and standard operating procedures for sample collection according to the possibility to process the samples within 4 or 24h and to the possibility to freeze the sample and transport it frozen. If transcriptomic analyses are required, RNAlater can be used, having been successfully used to preserve stool and saliva samples for transcriptomic analyses, although it may impact of DNA yield [163,164].

For large-scale epidemiological studies, samples collected during CRC screening for the fecal occult blood test have been used successfully, and no major degradation of bacterial DNA has been observed. Validation studies have shown that the collection kits kept at room temperature maintain stable results up to 14 days when compared to immediately frozen samples [165].

### 4.2. Microbiome Analysis

Even after the microbial genetic material has been extracted, there are many technologies and techniques available for sequencing and bioinformatic analysis, each with advantages and shortcomings (Table 2). Not only have we not reached standardization of methodology, but the human microbiome itself still remains partially unknown, with different levels of “dark matter” [166].

#### 4.2.1. Sequencing Techniques

The most widely used sequencing techniques to perform microbiome analysis and characterize community composition (taxonomic relative abundance) in human samples are high throughput 16S ribosomal RNA gene amplicon sequencing (16S rRNAseq) and whole shotgun metagenomics [166]. 16S rRNAseq is based on amplifying the 16S rRNA gene of bacteria by PCR before sequencing, allowing for a cheap characterization of the microbiome. Usually, a few variable regions of the 16S rRNA gene are sequenced (V3–V4), providing a resolution limited to the genus level [167]. The 16S rRNA can also be sequenced with long reads, expanding the whole gene, providing higher resolution [168]. Whole shotgun metagenomics, on the other hand, is based on sequencing the whole DNA present in samples and allows for the identification of species and genes of all microorganisms, not only of bacteria, provided that sequencing depth is adequate. In the case of 16S sequencing, the PCR amplification step guarantees that only microbial DNA will be sequenced. This is not the case for shotgun sequencing, where the DNA samples need to be enriched for microbial DNA beforehand.

#### 4.2.2. Bioinformatic Analysis

Bioinformatics analysis of 16S samples has traditionally relied on clustering similar sequencing reads up to a level of similarity (normally, 97%) into operational taxonomic units (OTUs). This clustering removes sequencing errors, but this also implies a loss of information. Alternatively, novel approaches attempt to retain all amplicon sequence variants (ASV). As opposed to clustering reads, they attempt to algorithmically distinguish sequencing errors from biological variation [169,170]. QIIME2 is a bioinformatics toolkit that provides frameworks for integrating all steps of 16S analysis [171].

In the case of shotgun metagenomics, many analysis lines are available. On one hand, read-based classification algorithms aim to provide a taxonomic assignment to each sequencing read. Taxonomic profiles allow us to analyze which microorganisms are present in biological samples, qualitatively and/or quantitatively. Different software implementations are available for this task [172]. Reads can also be classified by functional potential as opposed to taxonomy. These algorithms classify reads into gene families, which can provide different insights (identification of toxicity genes, reconstruction of pathways, etc.). In both cases, what can be detectable is limited to what is present in the databases that are used. Lastly, shotgun metagenomics reads can be used to reconstruct the original genomes (de novo assembly). This approach does not rely on any database, and thus it can be used to discover new genomes. For a review of bioinformatics methodology for shotgun metagenomics, please see the study by Breitwieser et al. [173].

#### 4.2.3. Statistics for Microbiome Analysis

Calculation of diversity metrics is common when analyzing taxonomic profiles. Diversity measures (Shannon, Simpson indices) are used to query the within-sample diversity or diversity, while β diversity metrics (Bray–Curtis, UniFrac) are used to investigate between-sample diversity. Besides diversity analyses, common statistics may be used to find statistically significant differences between groups. However, it is important to note that sequencing-derived microbiome datasets are compositional, that is, they do not provide absolute descriptions of the microbiome, but are relative to the whole microbiome present in each sample, requiring specific statistical methodology [174]. This complicates the interpretation of results and arises the possibility of spurious associations unless specific methodology is used. Alternatively, quantification of the total microbial load completely avoids the problem of compositionality and has been shown to provide more insights [175].

#### 4.2.4. Spatial In Situ Resolution

Although metagenomic and metatranscriptomic analyses have revolutionized the study of microbial communities, they have the main drawback of not providing spatial information on how these communities are distributed in the sample, thus preventing a full understanding of how the bacteria interact with each other or with the microenvironment [176,177]. Fluorescence in situ hybridization (FISH) targeting the rRNA can identify almost any microbe in a given tissue sample [178]. Fluorescence spectral imaging allows for the differentiation of many fluorophores identifying all members of a complex microbial community, thus offering a systems-level view of the spatial structure of the microbiome [179]. Because of technical limitations, rRNA FISH can only be used to differentiate only two or three microbial types simultaneously.

RNAscope is a recently developed RNA in situ hybridization technology that allows for direct visualization of RNA in formalin-fixed, paraffin-embedded (FFPE) tissue, enabling sensitive and specific spatial analysis of all RNA molecules present in a sample simultaneously [180]. In a study conducted by Serna et al., RNAscope technology was used to visualize *F. nucleatum* in rectal cancer tissue and to evaluate how this species [42] interacts with host cells within the tumor microenvironment. An automated version of the RNA in situ hybridization assay was originally developed for bacteria visualization in matched primary and metastatic CRC-intact FFPE tissues [94].

#### 4.2.5. Pre-Clinical Tools to Study Microbiome in Cancer

In vivo models are needed to understand the mechanisms through which some microbial communities or specific single microorganisms drive tumorigenesis [181]. Murine models provide excellent tools to study microbiota-associated human diseases [182]. Two main methods have emerged to explore the effects of the microbiota on physiology and disease in mice: germ-free models, which can be used for studying the functional properties of microbiome, and broad-spectrum antibiotic-treated models, which are used to study the cause–effect relationship between dysbiosis and resistance to therapies.

Beyond in vivo studies, in vitro models are also required to study the complexity of microbial interactions as they have the advantage of not being influenced by factors such as age, sex, diet, geography, genetic background, and antibiotic use, which may lead to bias in human and animal models [183]. Examples of in vitro models include organoids cultures and bioreactor system. The organoids are a three-dimensional culture of tissue that represent an excellent system for studying how microbiota induces and promotes cancer growth [184]. These technologies can be used to investigate the impact of dysbiosis on tumorigenesis and to find therapeutic strategies to modulate the microbiome to improve treatment efficacy [185]. For instance, the use of organoids in a study evaluating the role of *Helicobacter pylori* in gastric carcinogenesis was able to demonstrate how this species promoted cell proliferation and activation of the *c-Met* oncogene through NF-κB signaling [186].

Another in vitro model is the Bioreactor system, which allows for the study of complex gut microbial ecosystems in a controlled environment [187]. The “Robogut” bioreactor has been established in the Allen-Vercoe laboratory to culture gut microbial ecosystems in vitro under physiologically relevant conditions [188]. The laboratory uses the bioreactors as a model of the colonic microbiota in determining the effectiveness of antibiotic pretreatment in ulcerative colitis caused by *Clostridioides* [189]. The goal of this technology is to culture novel and highly fastidious species that cannot be cultured using conventional methods of cell culture in static dishes [190].

### 4.3. Challenges in Microbiome Studies in Cancer: Controlling for Bias

Observational studies of the intra-tumoral microbiome can be broadly grouped into (1) those with the goal of evaluating tissue microbiome composition in relation to prognostic events (treatment response/resistance, tumor recurrence, and tumor-related mortality) in patients with cancer or precursor lesions (case-only studies), and (2) those with the goal of comparing tissue microbiome composition between patients with cancer (or precursor lesions) and individuals free of cancer (case–control studies). A key component of group 1 studies is the evaluation of prognostic events over time. As in all clinical or epidemiological studies of the microbiome, other sources of microbiome variation need to be accounted for in the statistical analysis if they act as confounders or modifiers of the association between microbiome composition and prognostic events. Depending upon the tumor type, these could include clinical features of the diagnosed tumors (e.g., diagnostic method and stage, previous surgeries, familial gene mutations, other genetic variants, and diagnostic or prognostic biomarker levels), recent usage of pharmaceutical drugs (e.g., antibiotics, proton-pump inhibitors, metformin, and non-steroidal anti-inflammatories), demographic factors (e.g., place of residence, age, sex, and race/ethnicity), and lifestyle factors (e.g., diet or nutritional status at diagnosis, body weight, tobacco smoking, and alcohol consumption). Measurement errors in assessing lifestyle factors, especially dietary intake, must be carefully considered when planning a study. Nutritional status at diagnosis or surgery could be an alternative if an accurate dietary assessment by questionnaire is not available or feasible. Ideally, tissue specimens for microbiome analysis should be collected before therapeutic interventions, but this may not be possible if neoadjuvant chemo- or radiotherapy is indicated. In general, lifestyle and demographic factors have not shown strong associations with microbiome composition in terms of stool samples, but it should be noted that these associations remain largely unexplored in studies of tissue-specific or organ-specific microbiomes [191].

In group 2 studies, tissue microbiome composition is compared between cancer patients and cancer-free individuals—these are essentially case–control studies; moreover, in the absence of major biases due to study execution including recruitment strategy, microbiome analysis process, or differential errors in the measurement of epidemiologic variables, such studies would have the intention to evaluate tissue microbiome composition as a potential risk factor for the development of cancer [192]. In such studies, tissue samples collected for microbiome assessment are usually measured at, or shortly after, the date of diagnosis, and this fact is a major weakness of this study design since observed associations may not be causal. Further, depending upon the tumor site, the acquisition of normal tissues from cancer-free individuals from the same base population as the cases could range from moderately challenging to impossible. Normal colonic mucosal tissue from individuals free of cancer can be relatively easily obtained in studies using tissue collection via colonoscopy. Studies that utilize normal tissue obtained from national tissue banks or local tissue donor programs might be useful in giving a general overview of microbiome composition in individuals free of cancer, but caution should be exercised when interpreting differences with tumor tissues since tissue-bank or donor normal tissues may differ in other important ways such as age, overall health status, and other variables. The study of case–control differences in oral or gut microbiome composition from saliva or stool samples as proxies for tissue microbiomes or as risk factors themselves (e.g., via systemic effects on inflammation) has been considered in the majority of epidemiological studies of microbiome composition as a potential risk factor for cancer [22].

It is important to note that in the absence of a truly prospective cohort study in which biological samples for microbiome composition are collected before disease onset, microbiome-disease association signals observed in retrospective case–control studies may or may not be causal. Replication of observed association signals in additional case–control studies from similar and different populations is therefore necessary, as well as deeper mechanistic investigation through in vivo studies, for instance [191].

## 5. Future Directions

The study of microbiome as a new hallmark of cancer is just getting started. In this past year, 2048 publications related to “microbiome AND cancer” were indexed in PubMed, nearly 2000 more than 10 years ago. Whether understanding the tumor-associated microbiome will lead to a better comprehension of the pathogenesis of disease and corresponding molecular traits and will ultimately become a clinically useful biomarker tool for cancer prevention, diagnosis, and treatment is yet to be fully established, although evidence for this is beginning to accumulate. Examples of that are the microbiome-based screening tests for early detection of CRC, or the encouraging results of a phase I trial using FMT to boost ICI responses in refractory melanoma patients [131,193].

Despite the amount of knowledge being gathered, there are still some caveats that should be addressed. One of the most urgent is the standardization of microbiome methodology from sample collection to bioinformatic analysis in order to improve comparability/interpretation of results across studies [194]. Initiatives such as The Microbiome Quality Control (MBQC) project are already working to overcome this challenge. Special focus should be put on unveiling mechanistic processes to better define the link between microbiome (tumor-associated or from compartments distant from tumor-hosted organ) and carcinogenesis. More preclinical and clinical studies are needed to evaluate not only the community composition but also associated functional and multi-omic analyses. The Human Microbiome Project Consortium found shared metabolic pathways between healthy individuals despite having different microbiome taxa composition, which could also be the case in cancer patients [195]. Overall, there is a lack of longitudinal studies assessing the potential evolution of the microbiomes relevant for cancer. Both the microbiome and tumorigenesis are dynamic “systems”. Although viral and bacterial genomes appear to be stable in time in healthy individuals, point mutations in some bacteria could lead to a functional change, such as antibiotic resistance [195,196,197,198]. In addition, changes in extrinsic factors can also cause microbiome compositional variations over time and impact the results of microbiome manipulation strategies. Currently, different therapeutic strategies are under evaluation in clinical trials. Solving the knowledge gaps and the abovementioned weaknesses will allow clinicians to better determine who might benefit the most from these therapies. In fact, several questions remain to be answered regarding the use of microbiome therapeutics such as best approach or setting (in combination with standard chemo-, radio-, or immunotherapy, in metastatic or adjuvant settings), potential toxicities, ethical implications, and classification [3]. Of note, some of these therapies such as prebiotics or probiotics are widely used in the general population without proper regulation [199].

Microbiome research in oncology is an exciting field to be explored. The creation of collaborative multidisciplinary networks will be fundamental to augment the knowledge and optimize resources. Continued efforts should be made to overcome the challenges and ensure that we are ready for prime time.

## Figures and Tables

**Figure 1 ijms-22-01446-f001:**
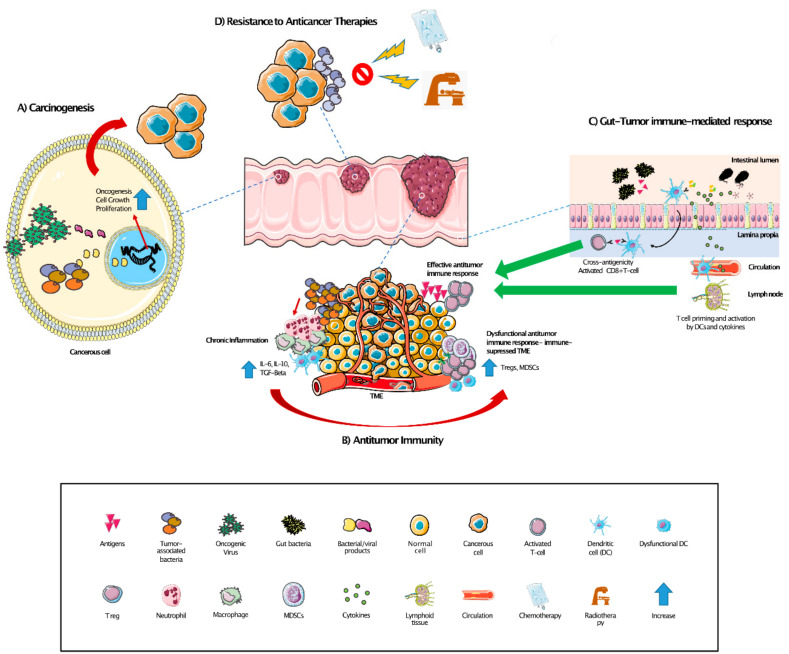
Impact of tumor-associated and gut microbiomes in cancer. (**A**) Carcinogenesis: intratumor bacteria and/or viruses and their by-products can activate oncogenic pathways and promote cell growth and proliferation. (**B**) Antitumor immunity: chronic inflammation caused by the local microbiome could lead to an immunosuppressive tumor microenvironment through altered antigen presentation and Tregs and myeloid-derived immunosuppressive cell (MDSC) stimulation, ultimately impairing anti-tumor immune-responses. (**C**) Gut–tumor immune-mediated response: gut bacteria and their by-products can enhance CD8+ T cell-mediated antitumor responses via (1) cross-reactivity of shared bacteria and tumor antigens recognized by T cells in the gut; (2) activation of dendritic cells, which will lead to T cell priming and expansion; (3) local pro-inflammatory cytokines or other bacterial products entering systemic circulation along with activated T cells. (**D**) Resistance to anticancer therapies: intratumoral bacteria can alter the efficacy of certain chemotherapies by altering the metabolism or through generating resistance to radiotherapy through hypoxic mechanisms.

**Figure 2 ijms-22-01446-f002:**
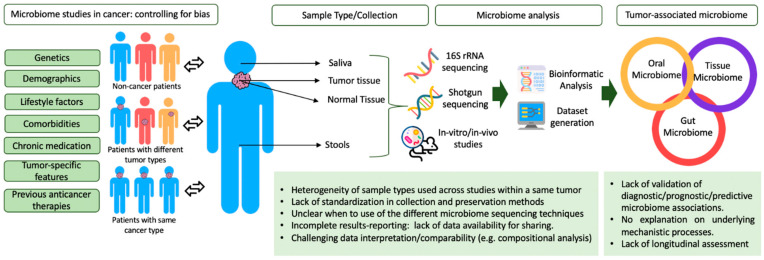
Challenges of microbiome studies in cancer.

**Table 1 ijms-22-01446-t001:** Tumor-associated microbiome.

Disease Site	Tumor Type	Sample Type	Tumor-Associated Taxa	Potential Clinical Utility Based on Recent Evidence
**Head and Neck and Upper Gastrointestinal Tumors**	Head and neck SCC	Saliva tumor tissue	*Kingella* and *Corynebacterium* [20,30]	Cancer prevention
			*Porphyromonas gingivalis* [31]	Diagnostic
			*Fusobacterium nucleatum* [32,33]	Prognostic
	Esophageal ADC	Tumor tissue	*Campylobacter* species [34,35,36]	Diagnostic
	Esophageal SCC	Tumor tissue	*Fusobacterium nucleatum* [13,33]	Prognostic
	Gastric carcinoma	Tumor tissue	*Helicobacter pylori* [6]	Diagnostic
**Hepatocarcinoma and** **Pancreatic Cancer**	Ductal ADC	Tumor tissue	VHB, VHC [5]	Screening Diagnostic
		Normal tissue	*H. pylori*, *P gingivalis*, *Fusobacterium* sp., *Aggregatibacter* sp., *Prevotella* sp., or *Capnocytophaga* sp. [37]	Diagnostic
	Pancreatic ductal ADC	Tumor tissue	*Pseudoxanthomonas* sp., *Streptomyces* sp., *Saccharopolyspora* sp., *Bacillus clausii, Proteobacteri* sp. [26]	Prognostic
			Gammaproteobacteria [38]	Predictive
**Colorectal Cancer**	Colorectal ADC	Tumor tissue Stool Saliva	*Fusobacterium nucleatum* [39,40,41,42,43]	Diagnostic Prognostic Predictive Therapeutic



		Tumor tissue	Enterotoxigenic *Bacteroides fragilis* [44]	Cancer Prevention
			*Escherichia coli (pk+)* [45]	
		Stool	*Peptostreptococcus stomatis, Parvimonas, Porphyromonas* [39,40] *Ascomycota, Basidiomycota, Orthobunyavirus* [46,47,48]	Diagnostic Screening tool
**Genitourinary tumors**	Urothelial carcinoma	Urine	*Fusobacterium,* Firmicute [49,50]	Diagnostic
	Renal cell carcinoma	Tumor tissue	*Chloroplast*, *Streptophyta* [51]	Diagnostic
	Prostate SDC	Tumor tissue	*Akkermansia muciniphila* [52]	Predictive of response
			*Listeria monocytogenes* [53]	Prognostic
	Endometrial cancer	Tumor tissue	*Porphyromonas* sp., *Atopobium vaginae* [54]	Diagnostic
**Lung cancer**	Lung ADC and SCC	Normal site	*Chlamydia pneumonia, Mycobacterium tuberculosi* [55]	Cancer prevention
		Saliva Tumor tissue	*Veillonella, Capnocytophaga, Selenomonas Megasphaera, Neisseria* [56]	Diagnostic
			Family Lachnospiraceae, genera Faecalibacterium and Ruminococcus [57]	Prognostic
		Faeces	*Akkermansia muciniphila* [14]	Predictive of response
**Breast cancer**	Triple-positive ductal ADC (HR/HER-2+)	Tumor tissue	*Bordetella*, *Campylobacter*, *Chlamydia*, *Chlamydophila*, *Legionella*, *Pasteurella* [58]	Diagnostic

	Triple-negative ductal ADC (HR/HER-2 -)	Tumor tissue	*Aerococcus*, *Arcobacter*, *Geobacillus*, *Orientia*, *Rothia* [58]	Diagnostic
**HPV-related cancers**	Oropharyngeal SCC	Saliva	*Lactobacillus*-enriched [32]	Diagnostic
	Cervical SCC	Tumor tissue	HPV16 [5]	Prognostic
		Vaginal fluid	*Lactobacillus*, *Gardnerella*, *Atopobium*, *Fusobacterium,* *Sneathia* [59]	Diagnostic Prognostic
**EBV-related cancers**	Nasopharyngeal carcinoma	Tumor tissue	EBV [60]	Diagnostic Prognostic
		Gut	Functional metabolic signature [61]	Prognostic

Abbreviations: SCC = squamous cell carcinoma; ADC = adenocarcinoma; HR = hormonal receptors; HPV = human papillomavirus; EBV = Epstein–Barr virus.

**Table 2 ijms-22-01446-t002:** Methodology for microbiome analysis: problems and solutions.

Type	Technique	Problem	Solution/Alternative
**Sequencing technique**	16S rRNA-seq	Low taxonomic resolution Limited functional analysis	Full-length 16S sequencing, shotgun sequencing
	Whole shotgun sequencing	More expensive Human DNA also gets sequenced	Sequencing at low coverage Adequate source material, enrichment of microbial material before sequencing
	Long read sequencing	Sequencing errors are difficult to detect	Combining long read sequencing with short read shotgun
**16S bioinformatics**	OTU-based methods	Loss of information in clustering	ASV-based methods
	ASV-based methods	Reliance on the algorithm to detect sequencing errors	
**Shotgun bioinformatics**	Taxonomic profiling	Reliance on incomplete databases	New assemblies will provide more complete databases
	Functional profiling	Reliance on incomplete databases, proteins of unknown function	Further characterization of microbial proteins is still needed
	De novo assembly	Incomplete assemblies, chimeric genomes, strain heterogeneity	Strict quality control Long-read sequencing will provide better assemblies
**Biostatistics**	Traditional statistics	Datasets are compositional	Compositional methods, estimation of total microbial presence to avoid compositionality
	Compositional analysis	Presence of zeroes Difficult to interpret	Zero-replacement
**Spatial in situ resolution**	RNA in situ hybridization	Low-throughput (only 2-3 bacterium can be detected)	Use it when information about spatial resolution is needed

## Data Availability

Not applicable.

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
