# Peer review of "Tumor-Associated Microbiome: Where Do We Stand?"

_ijms, 2021, doi:10.3390/ijms22031446_

Round 1
Reviewer 1 Report
Oliva and coworkers reviewed the recent progress on microbiome and cancer, which is an interesting and important field. The manuscript is well written. However, it is not well organized, there are arrangements and discussion to be done so that manuscript to be clearer. Please find below some comments/suggestions which might improve the quality of the manuscript.
- What do the numbers mean in ‘Microbiology meets Oncology (570)’ (line 44), ‘Tumor-Associated Microbiome (2600)’ (line 79), and ‘Microbiome and antitumor immunity (1000)’ (line 265)?
- The image resolution of figure 1 is not high enough. It is difficult to read the text in the figure.
- Only one figure is in this review. For the readers, more figures (describing some interesting work) or schematic (such as summarizing the challenges in microbiome studies in cancer) should be added.
- Refs should be added in the tables.
- Font: ‘A prognostic role of urine/tissue microbiome has not been described neither in urothelial carcinoma nor in kidney cancer.’ (line 216-217).
- Typo: ‘.’ (line 39).
Author Response
Dear reviewer,
thank you very much for your thorough review of the manuscript and your comments. Really appreciated it. See below the answers
- What do the numbers mean in ‘Microbiology meets Oncology (570)’ (line 44), ‘Tumor-Associated Microbiome (2600)’ (line 79), and ‘Microbiome and antitumor immunity (1000)’ (line 265)?
Answer: We apologize for this error. This was the word count for the section itself. We have removed this from the manuscript.
- The image resolution of figure 1 is not high enough. It is difficult to read the text in the figure.
Answer: We have augmented the resolution of the figure and adjusted the format to make it more readable. It is possible the quality is also reduced when uploading to the submission site as the figure needs to be attached to the main manuscript document. If still not able to see it well, I'ld be happy to send a TIFF file separately
- Only one figure is in this review. For the readers, more figures (describing some interesting work) or schematic (such as summarizing the challenges in microbiome studies in cancer) should be added.
Answer: We appreciate your suggestion. We have composed a new figure that graphically summarizes the process and current challenges microbiome studies in cancer patients. It is referenced in Section 4. (Figure 2)
- Refs should be added in the tables.
Answer: We have added the proper references in Table 1, which has also been re-formatted for clarity. Table 2 does not include references as it is not citing any author - it has been written by the authors.
- Font: ‘A prognostic role of urine/tissue microbiome has not been described neither in urothelial carcinoma nor in kidney cancer.’ (line 216-217).
Answer: We have corrected this.
- Typo: ‘.’ (line 39).
Answer: we have also corrected this. We have also reviewed the ma nuscript for other potential typos.
Thank you very much,
Marc Oliva

Reviewer 2 Report
This is an exhaustive and interesting review, discussing data reported so far on the potential role of microbiome in cancer. It is well written and faces also some important caveats and pitfalls, also froma technical point of view, that need to be afforded to clear the impact of microbiome in cancer promotion or rather in cancer immunosurveilliance.
Minor points.
1) Page 9, lane 158 and lane 170. There are two PMID (PMID 31383281 and 32962112, respectively) which probably refer to two references..maybe should be added to the list ? Similarly it happens at page 12, lane 260 (PMID 21483846). Finally, at page 18 lane 408, it appears this number: 31189463. Does it refer to anothe PMID, thus to another references?
2)Text should be formatted because often there are diferences in text size: i.e. Page 2 lane 45-60; page 7, lane 107; page 10 lanes193-193; page 11, lanes 216-217 and so on through all the text.
3) Page 20 lanes 447-452: some of the text is in bold characters
Author Response
Dear reviewer,
thank you very much for your thorough review of the manuscript and for your comments and overall feedback. They are really appreciated.
Please see the answers below to your comments.
Minor points.
1) Page 9, lane 158 and lane 170. There are two PMID (PMID 31383281 and 32962112, respectively) which probably refer to two references..maybe should be added to the list ? Similarly it happens at page 12, lane 260 (PMID 21483846). Finally, at page 18 lane 408, it appears this number: 31189463. Does it refer to anothe PMID, thus to another references?
Answer: We apologize for this mistake. You are right as these were references ion PMID format that were missed upon our last review and therefore were not included properly. We have already removed these PMIDs and added the references in the proper format.
2)Text should be formatted because often there are diferences in text size: i.e. Page 2 lane 45-60; page 7, lane 107; page 10 lanes193-193; page 11, lanes 216-217 and so on through all the text.
3) Page 20 lanes 447-452: some of the text is in bold characters
Answer for comment 2 ad 3: We have reviewed carefully the manuscript to correct the typos and the formatting errors.
Thank you very much again for your time in reviewing our work.
Best,
Marc Oliva

Round 2
Reviewer 1 Report
I recommend it for publication based on the serious revision.